# Identifying Consciousness in Other Creatures: Three Initial Steps

**DOI:** 10.3390/bs14040337

**Published:** 2024-04-17

**Authors:** Alejandro Heredia Cedillo, Dennis Lambert, Ezequiel Morsella

**Affiliations:** 1Department of Psychology, San Francisco State University, San Francisco, CA 94132, USA; dennis.lambert@ucsf.edu (D.L.); morsella@sfsu.edu (E.M.); 2Department of Neurology, University of California, San Francisco, CA 94158, USA

**Keywords:** consciousness, unconscious processing, evolution, encapsulation, passive frame theory

## Abstract

Identifying consciousness in other creatures, be they animals or exotic creatures that have yet to be discovered, remains a great scientific challenge. We delineate the first three steps that we think are necessary for identifying consciousness in other creatures. Step 1 is to define the particular kind of consciousness in which one is interested. Step 2 is to identify, in humans, the key differences between the brain processes that are associated with consciousness and the brain processes that are not associated with consciousness. For Step 2, to identify these differences, we focus on passive frame theory. Step 3 concerns how the insights derived from consciousness research on humans (e.g., concerning these differences) can be generalized to other creatures. We discuss the significance of examining how consciousness was fashioned by the process of evolution, a process that could be happenstance and replete with incessant tinkering, yielding adaptations that can be suboptimal and counterintuitive, far different in nature from our efficiently designed robotic systems. We conclude that the more that is understood about the differences between conscious processing and unconscious processing in humans, the easier it will be to identify consciousness in other creatures.

## 1. Introduction

The nature of consciousness, and how brains produce it, remains one of the greatest mysteries in science [1,2,3,4]. A related challenge is how to identify consciousness in other creatures [5], be they animals or exotic creatures that have not yet been discovered. Here, we attempt to delineate the first three steps that, in our estimation, are necessary for identifying consciousness in other creatures. 

Step 1 is to define the particular kind of consciousness in which one is interested. To this end, we are referring to basic, low-level consciousness (e.g., the subjective experience of pain, breathlessness, nausea, yellow afterimages, or ringing in one’s ears). This basic form of consciousness has been called ‘sentience’ [6], ‘subjective experience’, ‘phenomenal state’, and ‘qualia’ [7]. In this article, when we discuss consciousness, we are discussing this most basic form of consciousness, which is an experience of any kind. It might be ringing in the ears, a bright light, an afterimage, the smell of lavender, a song stuck in one’s head (an ‘ear worm’), or a phone number that one is trying to remember. A *conscious content* is any particular thing of which one is conscious (e.g., a percept, urge, or memory). The *conscious field* is composed of all that one is conscious of at one moment in time. 

When researchers speak about the puzzle of consciousness, and of how the brain produces it, they are referring to this most basic form of consciousness. This basic form of consciousness is contrasted with processes that are not associated with consciousness (e.g., the control of peristalsis or of the pupillary reflex). The empirical question is as follows: What is special about the brain circuitry that leads to conscious states, and why does the circuitry for peristalsis or the pupillary reflex, for example, not cause such states? 

Other forms of consciousness (e.g., the “narrative self” [8]), involve cognitive processes beyond that of basic consciousness. With this in mind, it is important to note that the mystery of basic consciousness (that is, how neural activity can give rise to an experience of any kind) would apply to conscious states that lack many of the sophisticated features that are often part of conscious states. The job of explaining basic consciousness is not the job of explaining these high-level features that are often, but not always, coupled with conscious states. Rather, the job is to explain how neural circuitry can give rise to an experience of any kind. For example, the mystery applies to conscious states that lack spatial extension, as in the case of some olfactory experiences. In addition, the mystery would apply to conscious states that are of extremely brief duration, as in the case of a hypothetical creature who lived for only a few seconds and was conscious during that span. The mystery applies to states that are utterly nonsensical, as in some hallucinations, anosognosia, and the dream world. Thus, for the mysterious phenomenon of consciousness to be instantiated, the phenomenon need not require (as far as we know) spatial extension, an extended duration, meaningfulness, or complexity regarding its contents. It might be that future research will reveal that one or more of these features is necessary for there to be a consciousness of any kind, but at present, as far as we know, it seems that these states could exist in some form, and be equally mysterious, when these features are absent. 

What is the difference between conscious and unconscious brain processes? Regarding this contrast, it is important to note that, in every field of inquiry within psychology and neuroscience, the contrast is made between conscious processes and unconscious processes. For example, in the field of perception, there exists the distinction between *supra*- versus *sub*liminal processing; in the field of memory, there is the distinction between ‘declarative’ (explicit) processes and ‘procedural’ (implicit) processes [9,10]. Similarly, in the field of motor control, the conscious aspects of voluntary behavior are contrasted with the unconscious aspects of motor programming [11], including the implicit learning of motor sequences [12] and the unconscious processes involved in speech production [13]. More generally, in many fields of study, there exists the contrast between ‘controlled’ processing, which tends to be associated with consciousness, and ‘automatic’ processing, which tends to be associated with unconscious mechanisms [14]. 

## 2. The Neural Correlates of Conscious Processing

Historically, consciousness has been associated with neural processes associated with perception (the “sensorium hypothesis”). Except for olfaction, most of these processes are housed in posterior areas of the brain. The association of consciousness with the sensorium stems from the observations that percepts are a common component of the conscious field and that motor programming (which muscle fibers should contract at which time to carry out an action) is believed to be largely unconscious [11].

It seems that the conscious processing of a mental representation (e.g., a percept) involves a wider and more diverse network of brain areas than does the subliminal (unconscious) processing of the same representation [15,16]. This evidence stemmed initially from research on anesthesia (e.g., [17]), perception [16], and unresponsive states (e.g., comatic or vegetative states [18]). Research in binocular rivalry suggests that some mode of interaction between widespread brain areas is important for consciousness [19,20]. Consistent with these views, actions that are consciously mediated involve more widespread activations in the brain than do similar but unconsciously mediated actions [21]. Accordingly, when actions are decoupled from consciousness (e.g., in neurological disorders), the actions often appear impulsive or inappropriate, as if they are not adequately influenced by all the kinds of information by which they should be influenced. One challenge in isolating the neural correlates of consciousness is that it is difficult to determine whether one is also including, in the neural correlates, activities that only support consciousness or the processes associated with the self-report of conscious contents [22]. 

Regarding the neuroanatomy responsible for consciousness, investigators have focused on vision. In vision research, controversy remains regarding whether consciousness depends on higher-order perceptual regions [23,24] or lower-order regions [25,26]. Some theorists have proposed that, while the cortex may elaborate the contents of consciousness, consciousness is primarily a function of subcortical structures [27,28]. Regarding the frontal lobes, it has been proposed that, although they are involved in cognitive control, they may not be essential for the generation of basic consciousness [27,29]. This hypothesis is based in part on research on anencephaly [27], the disruption of cortical functions (e.g., through direct brain stimulation or ablation [28]), and the psychophysiology of dream consciousness, which involves prefrontal deactivations [30]. The surgical procedure of frontal lobotomy, once a common neurosurgical intervention, was never reported to render patients incapable of sustaining consciousness. However, research on patients with profound disorders of consciousness (e.g., vegetative state) suggest that signals from the frontal lobes may be essential for the instantiation of consciousness [31,32,33].

Consistent with the sensorium hypothesis, evidence suggests that perceptual regions are the primary site of consciousness. For example, electrical stimulation of parietal areas gives rise to the conscious urge to perform an action, with increased activation making subjects hallucinate that they executed the action, even though no action was performed [34,35]. In contrast, activating motor areas (e.g., premotor regions) leads to the expression of the actual action, but subjects believe that they did not perform any action whatsoever (see also [36]). The urge to perform a motor act is associated with activation of perceptual regions. Consistent with the sensorium hypothesis, the majority of studies involving brain stimulation and consciousness have found that stimulations of perceptual (e.g., posterior) brain areas lead to changes in consciousness (e.g., haptic hallucinations). In the literature, we found only one piece of evidence [36,37] in which brain stimulation of a frontal area led to conscious content. In the study, weak electrical stimulation of the pre-supplementary motor area led to the experience of the urge to move a body part, with stronger stimulation leading to movement of the same body part. It has been proposed that such activation led to feedback (e.g., corollary discharge) that is then ‘perceived’ by perceptual areas [38]. This finding is, thus, consistent with the sensorium hypothesis. 

Consistent with the sensorium hypothesis, a key component of the control of intentional action is feedback about ongoing action plans to perceptual areas of the brain, such as the post-central cortex [34]. With this in mind, it has been proposed that consciousness is associated not with frontal or higher-order perceptual areas but with lower-order perceptual areas [1]. However, it is important to qualify that, though the sensorium hypothesis specifies that consciousness involves neural circuits that, traditionally, have been associated with perception, such circuits are widespread throughout the brain and exist within both cortical and subcortical regions [39]. 

Several features of the olfactory system render it a fruitful modality in which to isolate the substrates of consciousness. For example, olfaction can reveal much about the role of thalamic nuclei in the generation of consciousness: Unlike most senses, afferents from the olfactory sensory system bypass the first-order relay thalamus and directly target the cortex ipsilaterally [40]. Second, one can draw some conclusions about second-order thalamic relays (e.g., the mediodorsal thalamic nucleus; MDNT). After cortical processing, the MDNT receives inputs from olfactory cortical regions [41]. The MDNT plays a significant role in olfactory discrimination [42], olfactory identification, and olfactory hedonics [43], as well as in more general cognitive processes including memory [44], learning [45], and attentional processes [46,47]. However, there is no evidence that a lack of olfactory consciousness results from lesions of any kind to the MDNT (see discussions about this possibility in [48]). 

Consistent with ‘cortical’ theories of consciousness, complete anosmia (the loss of the sense of smell) was observed in a patient with a lesion to the left orbital gyrus of the frontal lobe [49]. In addition, a patient with a right orbitofrontal cortex (OFC) lesion experienced complete anosmia [50], suggesting that the OFC is necessary for olfactory consciousness. In addition, conscious aspects of odor discrimination have been attributed to the activities of the frontal and orbitofrontal cortices [51]. Keller [52] concludes, “There are reasons to assume that the phenomenal neural correlate of olfactory consciousness is found in the neocortical orbitofrontal cortex” (p. 6; see also [53]).

Step 2 is to identify, in humans, the key differences between the brain processes that are associated with consciousness and the brain processes that are not associated with consciousness. For Step 2, an acceptable answer to the question regarding the difference between conscious and unconscious processes cannot be that all physical things are conscious or that all neural activities are conscious, because it is posited in the question itself that, in humans, there exist unconscious processes. Hence, for Step 2, one must identify and explain the difference between conscious and unconscious processes as they exist in the human brain (not in a computer, plant, or robot).

Regarding the answer to the question posed in Step 2, the difference between conscious and unconscious processes is not as obvious as one might suppose. For example, unconscious processes are capable of both stimulus detection and issuing motor responses to stimuli, at least under some circumstances. Moreover, it is not the case that unconscious processes are less sophisticated than conscious ones. There are many examples that reveal the sophistication of unconscious processes, for example, as in the case of syntax, motor programming, and unconscious inter-sensory processing, each of which is, to a large extent, unconsciously mediated. 

Once one has identified the difference between conscious and unconscious processes in humans, one should, ideally, explain how this difference accounts for the phenomenological data, including the peculiarities of individual conscious contents. 

According to one theoretical framework, the Projective Consciousness Model [54], a potential constitutive marker of consciousness (in humans, at least) is the ability of the brain to create a first-person perspectival viewpoint situated within a three-dimensional spatial environment, a format that can be understood through the principles of projective geometry. Notably, such a viewpoint is experienced both during waking and dreaming. Some such advance in defining what constitutes consciousness is needed in order to move beyond tautological formulations such as the often quoted, “something it is like” [55], which actually presupposes “consciously like” (we thank Björn Merker for permission to quote this observation from an unpublished manuscript of his). It is interesting to consider whether the Projective Consciousness Model can be applied not only to visual consciousness but to olfactory consciousness, which lacks the spatial extension and dimensions of visual consciousness but does possess the separation between the conscious content (the odorant) and the observing agent [56,57,58].

Another framework that attempts to identify the difference between conscious and unconscious processes is passive frame theory (PFT [1]). In this article, we will focus on this framework for three reasons. First, PFT was developed in our laboratory and is the framework with which we are the most familiar. Hence, it is the framework with which we are most able to make inferences regarding the presence of consciousness in other creatures. Second, in line with the basic form of consciousness defined in Step 1, of all the proposed functions of consciousness, the function proposed by PFT happens to be the most basic, low-level, function. Third, pertinent to the aim of discovering the “behavioral correlates of consciousness” (the BCCs [5]), this low-level function involves overt behavior. This last property of the framework is necessary for the thought experiment presented below in Step 3. 

PFT is a synthesis of a diverse group of action-based approaches to the study of consciousness and of perception-and-action coupling. According to PFT, the conscious field is necessary to integrate specific kinds of information for the benefit of *action selection*, a stage of processing that is neither perceptual nor motor. Action selection, as when one presses one button versus another button or moves leftward versus rightward, is distinct from motor control/motor programming processes [59], which are largely unconscious. In the functional architecture proposed by PFT, integrations and conflicts occurring at perceptual stages of processing (e.g., the ventriloquism effect and the McGurk effect [60]) can occur unconsciously, as can conflicts or integrations occurring at stages of processing involving motor programming (e.g., coordinating the specifications of articulatory gestures [61]) or smooth muscle processing (e.g., the pupillary reflex and peristalsis). The conscious field is unnecessary for such integrations and the resolution of such conflicts (e.g., in the McGurk effect) in these stages of processing. 

In contrast, conflicts occurring at the processing stage of action selection must perturb the conscious field. For example, when one endures pain, holds one’s breath while underwater, or suppresses elimination behaviors, one is conscious of the inclinations to perform certain actions and of the inclinations to not perform those very same actions. Such a scenario is a *conscious conflict*. That is, mental representations that are associated with simultaneously activated, conflicting action plans are experienced as phenomenal states. In PFT, the conscious field benefits action selection by permitting the “collective influence” of these conflicting inclinations toward the *skeletomotor output system*, as captured by the principle of Parallel Responses into Skeletal Muscle (PRISM). Such a collective influence of conflicting inclinations, existing only in virtue of the conscious field, occurs when withstanding pain, holding one’s breath while underwater, or suppressing elimination behaviors. 

When action control is decoupled from consciousness, as occurs in some neurological disorders (e.g., automatisms and in anarchic hand syndrome), action selection can no longer reflect collective influence (e.g., of conflicting inclinations). In such cases, actions appear impulsive and not influenced by all the kinds of information by which context-sensitive actions should be influenced. Although these unconsciously-mediated actions can be quite sophisticated (e.g., the manipulation of tools [62]), they are “un-integrated”, that is, they are not influenced by relevant contextual factors.

## 3. Encapsulation

Why is the conscious field often composed of conflicting inclinations? From the standpoint of PFT, this is because most of the contents that arise in the conscious field are “encapsulated” from other contents (including inclinations) and from voluntary processing. A conscious content is said to be encapsulated when its nature or occurrence cannot be influenced directly by voluntary processing or by higher-level knowledge, as in the case of perceptual illusions [63,64]. Sacks [65] refers to such conscious contents as “autonomous”, because one cannot voluntarily influence their nature or occurrence. There are countless cases in which one’s desires cannot override the nature of encapsulated conscious contents, as is obvious in the case of physical pain or nausea. Regarding perceptual illusions (e.g., the Müller-Lyer illusion), these illusions persist despite one’s knowledge regarding the true nature of the perceptual stimulus. The nature of encapsulated conscious contents has been investigated using the reflexive imagery task (RIT). In the RIT (see review in [66]), participants are instructed not to perform a certain mental operation in response to external stimuli. For example, before being presented with a line drawing of a cat, subjects might be instructed not to think of the name of the to-be-presented visual object. Despite the intentions of the subject, the undesired mental operations often arise, yielding, for example, “cat” (i.e., /k/, /œ/, and /t/) for the stimulus CAT. Substantive RIT effects (occurring in roughly half the trials) arise even when the visual object (e.g., CAT) is presented only in the periphery, and participants are focused on a separate task (e.g., the flanker task [67]). RIT effects arise even though the involuntary effect involved processes as sophisticated as word-manipulations, as occur in the childhood game of Pig Latin; syntactic processing; and mental arithmetic (see review of evidence in [66]). What determines what is activated in consciousness in the RIT? It seems that the strength of the association between the stimulus and a particular mental representation is determinant. In one RIT [68], participants first learned to associate pseudowords with nonsense objects (*n* = 6). One of the words was shown with the object in sixty trials, creating “strong associates”, while the other word was shown with the object in only ten or twenty trials, creating “weak associates”. Participants were later presented with each object and instructed not to think of any of the associated pseudowords. Strong associates (proportion of trials: *M* = 0.52, *SD* = 0.40) were more likely to enter consciousness than weak associates (*M* = 0.21, *SD* = 0.32, *F* (19) = 2.51, *p* = 0.021 (Cohen’s *d* = 0.56).

In PFT, the contents of consciousness (e.g., nausea, the experience of a yellow afterimage, or the smell of lavender) are independent from one another, stemming from content generators that are themselves independent of each other. This is analogous to how, at certain stages of music production, the various “tracks” from a recording console at a recording studio are independent from one another [69]. When one listens to a song on the radio, it is not obvious to one that each instrument composing the music heard in a song is actually recorded separately on an independent “track.” Once the many tracks composing a song are recorded, the mixing console is used to collate (“mix”) the tracks and generate one master recording that, when played on the radio or through some other medium, presents to the listener all the instruments playing simultaneously. In this analogy, the song one hears on the radio, which is actually the integrated, co-presentation of all the independent tracks, would be analogous to the conscious field, which presents the unified collation of all the conscious contents activated at one time.

According to PFT, most of the contents of consciousness are, and should be, encapsulated, not only from voluntary processing but from each other, such that one conscious content cannot directly influence the nature or occurrence of another conscious content. The fact that encapsulation is adaptive during ontogeny is obvious when one considers the negative consequences that would arise if a young child could, by an act of will, directly control the occurrence of pain, fear, or guilt [70].

It is only through the conscious field that these encapsulated contents can influence action selection collectively. Figuratively speaking, the conscious field is a mosaic composed of the collation of all the encapsulated contents that are activated at one moment in time. The conscious field is sampled only by the (unconscious) response codes of the skeletomotor output system. Unlike the ‘workspace’ models associated with the *integration consensus* (e.g., [71,72]), in which conscious representations are ‘broadcast’ to modules engaged in both stimulus interpretation and content generation, in PFT (as in [73]), the contents of the conscious field are directed only at (unconscious) response modules in the skeletomotor output system.

From the standpoint of PFT, the conscious field evolved as an adaptation that allows conscious contents, each of which is encapsulated, to influence action selection collectively. For example, the conscious content of a cupcake activates the desire to eat it. This urge cannot be easily turned off at will. However, the conscious field is simultaneously populated by the conscious content of, for example, the doctor reminding one to cut down on sweets. Because of the conscious field, action selection is influenced by both contents. The field simply permits these conscious contents to influence action selection simultaneously (“collective influence”). Figuratively speaking, each conscious content “does not know” of the nature of the other conscious contents and its nature is not influenced by the nature of the other contents. The field operates blindly, whether there is conflict between conscious contents or not. In this sense, it is passive.

Without the field, however, there would not be the collective influence of contents when conflicts arise. When the conscious field is absent or not operating properly, action selection arises but is not adaptive, revealing a lack of collective influence: Actions are not guided by all the kinds of information by which they should be guided. For example, one eats the cupcake without taking into account what the doctor recommended. The conscious field permits what has been called *conditional discrimination*, a kind of contextualized response in which one responds to stimulus *X* in light of stimulus *Y* [73,74,75,76].

We should add that, from this standpoint, a conflict is not necessary for the conscious field to operate, but the conscious field operates incessantly in order to deal with conflicts when they arise. In some ways this resembles other biological systems. For example, the kidneys filter the blood regardless of whether the blood needs filtering (similar to a pool filter).

The architecture proposed by PFT, one in which encapsulated conscious contents blindly activate their response codes, leads to the following question. Why is the conscious field necessary if each content could just activate its corresponding response code (*R*) and then the differential activation of *R*s could determine action selection, without any need for a conscious field? Our first response to this question is that PFT is a *descriptive* account of brain function, describing these processes as they evolved to be. It is not a *normative* account, which describes processes as they ought to be designed. Thus, our first response is that, for some reason, Mother Nature solved the problem of collective influence by using the conscious field. Hypothetically, perhaps this problem could have been solved in other (unconscious) ways, as in the case of determining action selection by simply summing up the relative strength of the *R*s. Our second response to this question is that, today, it is known that, considering the number of stimuli that are presented usually to the visual system alone, conditional discriminations (e.g., responding to stimulus *X* in light of other stimuli at that moment [74,75,76]) can become computationally impossible [77,78].

In other words, responding adaptively to one stimulus in the context of all the other stimuli in the conscious field is unsolvable through formal computational means. It is important to emphasize that this is so even when considering only visual stimuli. In everyday life, this problem is compounded by the fact that the conscious field contains contents from many more modalities than just vision, and the field is populated as well by non-sensory elements (e.g., urges, linguistic representations, and memories), all of which can influence action selection. In collective influence, the response to a stimulus is influenced by all the contents in the field. It is a response to the configuration of all stimuli.

To complicate matters, in some circumstances, the adaptive response to Stimulus *X* depends on the spatial distance between Stimulus *Y* and Stimulus *Z*, as occurs in soccer, for example. Whether a defender passes the ball back to the goalie can depend on the distance between a midfielder and a member of the rival team. Such circumstances are often unpredictable, thus requiring the conscious field to always represent the spatial relations amongst all the conscious contents that are activated at a given time. This example reveals that action selection should not be determined just by the relative strength of activation of the response codes activated by each stimulus in a stimulus array. Action selection must also be influenced by the relations among the stimuli. In ways that have yet to be identified, the conscious field achieves this while formal computational systems cannot.

According to PFT, the problem of conditional discriminations and collective influence is solved, somehow, by the conscious field. The solution might be very different in nature to what engineers would design to solve this problem in a robotic system. A good analogy is found in an audio speaker. Because of its structure, configuration, and properties, an audio speaker can simultaneously reproduce, for example, all the notes that are played by a band. Let us say that the band consists of a guitarist, a bassist, and a flutist. The speaker can oscillate and reproduce continuously and simultaneously the different frequencies produced by the three musicians. At one moment, the nature of the vibration of the speaker reflects, for example, notes from both the bass guitar and the flute. It does so blindly and without any computation. It achieves this because of the way in which sine waves happen to blindly interact with each other. If one were to achieve this through computer programming, it would be an extremely complicated affair. But no such programming is needed because of the way in which sine waves happen to interact. The physical nature of the speaker, in a sense, solves a problem that would be difficult to solve through traditional computational means. Similarly, we propose that the conscious field is a trick by which Mother Nature solved the problem of collective influence and conditional discriminations. This trick might be different from the way an engineer might solve the problem.

In summary, according to PFT, consciousness is for what in everyday life is called “voluntary” action. The theory proposes that consciousness serves as an information space that passively enables flexible, context-sensitive action selection, yielding integrated action. The theory also reveals that the role of consciousness is very circumscribed, serving only the skeletal muscle output system, which, in turn, subserves the somatic nervous system (Figure 1). (The somatic nervous system, which innervates skeletal muscle and relays somatic (external) sensory information, is often contrasted with the autonomic nervous system, which innervates smooth muscle, cardiac muscles and glands, and relays visceral (internal) sensory information).

Step 3 concerns how the insights derived from consciousness research on humans can be generalized to other creatures. For Step 3, one must first consider that the products of evolution are not designed the way we humans may think they ought to be designed. Over eons, consciousness was fashioned by the process of evolution, a process that could be happenstance and replete with incessant modifications [79]. Hence, the products of evolution can be suboptimal and counterintuitive [79,80,81,82,83]. Such products could be far different in nature from those of robotics [84], which involve, for example, designing optimized solutions for pre-specified behavioral outcomes.

Consider two examples: the artificial heart is very different from its natural counterpart, and the difference between human locomotion and artificial locomotion is a stark one—that between legs versus wheels. Thus, when *reverse engineering* biological products, the roboticist cautions, “Biological systems bring a large amount of evolutionary baggage unnecessary to support intelligent behavior in their silicon-based counterparts” (p. 32, [84]). Similarly, the ethologist concludes, “To the biologist who knows the ways in which selection works and who is also aware of its limitations it is no way surprising to find, in its constructions, some details which are unnecessary or even detrimental to survival” (p. 260, [81]). With this in mind, when reverse engineering the products of nature, the student of consciousness must abandon a normative view (which describes how things should function) and adopt instead a more humble descriptive view (which describes the products of nature as they have evolved to be; see a theoretical account of the evolution of the nervous system in [85]).

Second, one must consider that, across different creatures, the same behavioral capacity, or cognitive function, can be carried out by vastly different mechanisms. Hence, the proposal that creature *X* must possess consciousness because it features a certain behavioral capacity requires that several assumptions be made. In the natural world, it is not always the case that the same behavioral capacity (e.g., navigating toward a stimulus) stems from the same underlying mechanism (e.g., vision versus echolocation). Moreover, even within one species (e.g., humans), the same behavioral operation (e.g., the closing of the eyelids) could be carried out by more than one kind of mechanism. Consider the difference between an eye blink (which normally occurs automatically and does not require conscious processing) and a wink (which is a high-level action that requires conscious processing and falls under the rubric of “voluntary” action). To an observer, these two actions might seem indistinguishable, even though their underlying processes are very different—to the actor, these two actions certainly feel very different. Consider also that the constriction or dilation of the pupils is, to an observer, an observable act. Unlike a blink or a wink, however, pupil dilation is carried out by smooth muscle effectors, in response to various environmental and internal conditions (e.g., ambient lighting and emotional arousal). People are not normally aware of such actions involving smooth muscle, yet these actions transpire quite often. To take another example, consider that a simple smile can emerge from multiple, distinct cognitive and neural mechanisms, depending on whether it is a genuine smile or a forced smile [86]. Thus, even within the same organism and with the same effector, the same kind of behavioral capacity can be implemented via vastly different mechanisms [87], some of which may be more elegant and efficient than others. Nature does things differently, for better or worse. The job of the scientist is to describe the products of nature as they are and as they function.

Inferring that creature *X* must possess consciousness because it features a certain behavioral capacity that, in humans, is associated only with consciousness is most compelling when that creature shares a common ancestor with humans. For such creatures, that capacity would be *homologous*. If there is no such shared ancestry, then the argument would be far less compelling, for, as mentioned above, it could be that the same behavioral capacity (e.g., navigating toward a stimulus) stems from different underlying mechanisms (e.g., vision versus echolocation). Thus, if the behavioral capacity is *analogous* (e.g., flight in birds versus in bats), then more assumptions must be made, and additional convergent evidence must be presented, when proposing that creature *X* possesses consciousness.

It is important to not equate consciousness with intelligence. From the standpoint of PFT, the conscious field is one of many adaptations in the animal kingdom that leads to intelligent behavior. Many processes associated with intelligent behavior arise from neural processes and structures unassociated with consciousness (e.g., in the cerebellum). Many of the unconscious processes in the nervous system (e.g., peristalsis and the pupillary reflex) could be said to be intelligent. There are many forms of intelligence in artificial systems and in the animal kingdom (e.g., reflexes in a sea slug) which stem from structures that are not homologous to those associated with human cognition and the conscious field. It is a mistake to assume that an intelligent process (e.g., in a robotic system) must stem from a conscious process. At this stage of understanding, it is important to take into consideration whether the intelligent structure is homologous with those associated with consciousness in humans.

With the above in mind, if one subscribed to PFT and encountered an exotic creature that displayed the ability to respond to a stimulus in a contextually sensitive manner that resembled the “integrated” actions of humans, this by itself would not imply that that creature possesses a conscious field. That is, the discovery of a BCC for an exotic creature could be construed as a necessary yet not sufficient criterion by which to infer consciousness in that creature. The creature’s ability, which could be analogous to that of humans, could be due to something other than a conscious field, perhaps to mechanisms that are, as of yet, unknown to us. Additional convergent evidence, including about the physical substrates of that ability in that creature, would be necessary to warrant the conclusion that the creature possesses a form of consciousness.

However, such a behavioral capacity in the chimpanzee would strongly suggest the presence of a conscious field, because, in terms of parsimony and phylogeny, that ability is more likely to be homologous to the human ability than to be only analogous to the human ability. This is because evolutionary changes (e.g., divergent and convergent evolution) emerge, largely, from interactions among environmental factors and incremental modifications of developmental processes and regulatory mechanisms [88,89]. Moreover, fundamental structural and functional organizational principles (e.g., cochlear mechanisms [90] and auditory processing [91]) are highly conserved across the taxa comprising a monophyletic group. Thus, within a phylogenetic branch, it is more likely for a specific adaptation (e.g., echolocation) to have evolved only once rather than more than once [92]. For example, in cetaceans, it is more likely the case that the echolocation abilities of the beluga whale and the sperm whale are homologous, i.e., were inherited from a common ancestor, rather than analogous, i.e., that echolocation abilities evolved independently in each of the species.

But even when generalizing consciousness to the chimpanzee, our close phylogenetic relative, one must remain conservative. Consider that the hippopotamus and the beluga whale, though closely related and sharing a common ancestor on the phylogenetic tree, possess different cognitive abilities and morphologies (e.g., legs versus flippers). Another example would be that, though genetically related, humans possess language but chimpanzees do not and that both humans and chimpanzees do not share the echolocation abilities of their fellow intelligent mammal, the beluga whale.

## 4. Conclusions

The more that is understood about the nature of consciousness in humans, including the differences between conscious processing and unconscious processing, the easier it will be to identify consciousness in other creatures, be they animals or exotic creatures yet to be discovered. Here, we introduced only what we consider to be the first three steps that are necessary for such an identification. The myriad, varied, and counterintuitive solutions provided by Mother Nature, when combined with the observations noted in Step 3, lead us to conclude that, when attempting to identify consciousness in other creatures, we will encounter many interesting surprises, ones that will challenge our assumptions about the origin(s) and evolution of intelligent beings everywhere.

## Figures and Tables

**Figure 1 behavsci-14-00337-f001:**
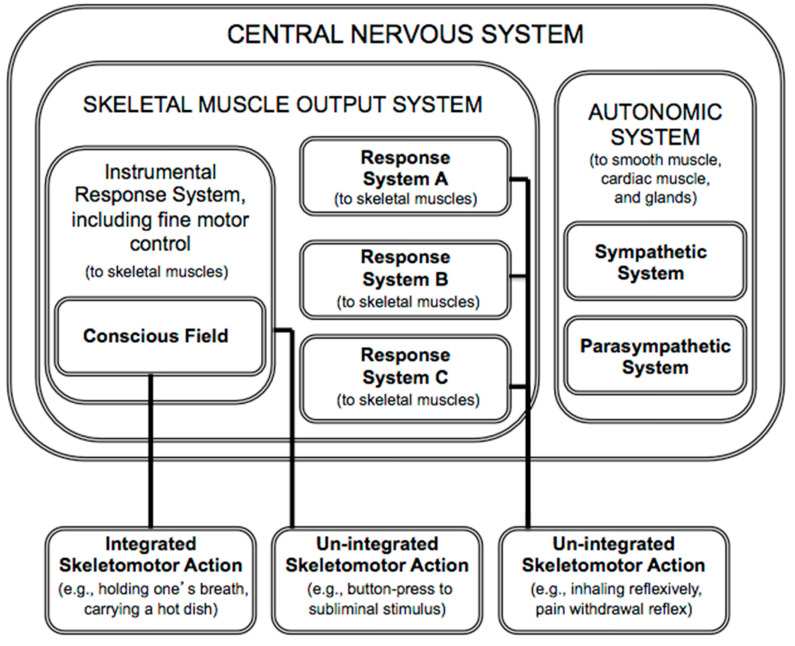
The circumscribed domain of consciousness within the nervous system (based on [1]). Response systems can influence action directly, as in the case of “un-integrated” actions. It is only through the conscious field that multiple response systems can influence action collectively, as when one holds one’s breath while underwater (a case of “integrated” action).

## Data Availability

Not applicable.

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
