# Peer review of "Identifying Consciousness in Other Creatures: Three Initial Steps"

_behavsci, 2024, doi:10.3390/bs14040337_

Round 1

Reviewer 1 Report

Comments and Suggestions for Authors

Author Response

See attached letter

Reviewer 2 Report

Comments and Suggestions for Authors

The problem tackled by the authors is the consciousness ascription problem. How do we identify consciousness or its absence in other beings? It is a widely discussed problem and has not been solved as yet, nor have many meaningful strategies to the problem been articulated. Because of this, it is not surprising the present authors are also not able to solve the problem. They side step this by outlining the first 3 steps to the problem, as they see them. This reader wasn't able, however, to feel any closer to dealing with the ascription problem after reading this article.

However, I feel the same way after reading all articles on this topic, except those that focus on just how hard, perhaps impossible, the ascription problem is. That of course is this reader's take on the problem and that doesn't mean the present authors should not be able to have their say on what their approach to the problem is. But the authors do need to re-work their piece to some extent to deal with some of its shortcomings.

The biggest problem is in step 2 in which there is to be a distinction made between neural process constitutive of consciousness and those which are not. However, this is a key crux of the ascription problem. How are we to do that exactly? There's a whole literature on this and this literature does not get discussed at all.

Another problem is that step 1 is about defining the sense of consciousness being referred to but despite this I found myself not really knowing exactly which sense the authors were referring to in different places of the article. Is it phenomenal consciousness? Is it higher-order or access consciousness? Is it voluntary decision-making effecting the skeletal system? I found this to not be clear. And if the latter, along with identifying behavioral correlates of consciousness, well that seems to me to just beg the ascription question all over again.

That is, surely a decision-making neural mechanism could be identified in a bee that would satisfy a voluntary decision description and then the authors might say, there that's consciousness and the bee is conscious. Really though? I would be left saying but that's a set of neurons making a 'decision' to effect behavior. I have no idea if that 'decision' is accompanied by or constitutes consciousness. So what progress have we made?

While I understand that the authors might see the point of their paper as framing the ascription problem within frameworks with which they are familiar, which is fine and reasonable, I am not led through those efforts to be any the wiser on just how we will overcome the ascription problem via this frameworking. The challenge to the authors is to convince this reader, and no doubt others, how their frameworking might overcome, or at least in some way contribute to shedding light on, the usual problems discussed in the literature, rather than sidestepping such problems altogether. At the very least the usual problems need to be better discussed.

Fig 1 is poorly explained.

Typo line 288, 'conscious' should be 'consciousness'

Author Response

Please see attached letter

Round 2

Reviewer 2 Report

Comments and Suggestions for Authors

Point 1 in particular has been inadequately dealt with. I don't see how addition of that paragraph does anything to address the literature on distinguishing consciousness-relevant neural correlates of consciousness (i.e., the constitution of consciousness) and consciousness-irrelevant neural correlates of consciousness (called various other things in the literature). If the authors don't know what this literature is, it shows they need to read more in this area if they are to set out their first step as they do.

For the authors' response to point 2, your short paper is not a 'treatise'. The response doesn't really get to my point either in that if that is the type of consciousness you refer to, why does much of your paper seem to look like you are talking about access consciousness of the sort in voluntary decision-making. Maybe I'm just not getting it, but other readers could also therefore struggle to get it.

I'm not sure the response to point 4 does anything to discuss the usual problems discussed in consciousness ascription literature.

Check for typos e.g. ''collective influenced"
